# LDMIC: Learning-based Distributed Multi-view Image Coding

**Xinjie Zhang, Jiawei Shao, Jun Zhang**
The Hong Kong University of Science and Technology, Hong Kong, China
{xinjie.zhang, jiawei.shao}@connect.ust.hk, eejzhang@ust.hk

## Abstract

Multi-view image compression plays a critical role in 3D-related applications. Existing methods adopt a predictive coding architecture, which requires joint encoding to compress the corresponding disparity as well as residual information. This demands collaboration among cameras and enforces the epipolar geometric constraint between different views, which makes it challenging to deploy these methods in distributed camera systems with randomly overlapping fields of view. Meanwhile, distributed source coding theory indicates that efficient data compression of correlated sources can be achieved by independent encoding and joint decoding, which motivates us to design a learning-based distributed multi-view image coding (LDMIC) framework. With independent encoders, LDMIC introduces a simple yet effective joint context transfer module based on the cross-attention mechanism at the decoder to effectively capture the global inter-view correlations, which is insensitive to the geometric relationships between images. Experimental results show that LDMIC significantly outperforms both traditional and learning-based MIC methods while enjoying fast encoding speed. Code is released at https://github.com/Xinjie-Q/LDMIC.

## 1 Introduction

Multi-view image coding (MIC) aims to jointly compress a set of correlated images captured from different viewpoints, which is promising to achieve high coding efficiency for the whole image set by exploiting inter-image correlation. It plays an important role in many applications, such as autonomous driving (Yin et al., 2020), virtual reality (Fehn, 2004), and robot navigation (Sanchez-Rodriguez & Aceves-Lopez, 2018). As shown in Figure 1(a), existing multi-view coding standards, e.g., H.264-based MVC (Vetro et al., 2011) and H.265-based MV-HEVC (Tech et al., 2015), adopt a joint coding architecture to compress different views. Specifically, they follow the predictive compression procedure of video standards, in which a selected base view is compressed by single image coding. When compressing the dependent view, both the disparity estimation and compensation are employed at the encoder to generate the predicted image. Then the disparity information as well as residual errors between the input and predicted image are compressed and passed to the decoder. In this way, the inner relationship between different views decreases in sequel. These methods depend on hand-crafted modules, which prevents the whole compression system from enjoying the benefits of end-to-end optimization.

Inspired by the great success of learning-based single image compression (Ballé et al., 2017; 2018; Minnen et al., 2018; Cheng et al., 2020), several recent works have investigated the application of deep learning techniques to stereo image coding, a special case of MIC. In particular, Liu et al. (2019), Deng et al. (2021) and Wödlinger et al. (2022), mimicking traditional MIC techniques, adopt a unidirectional coding mechanism and explicitly utilize the disparity compensation prediction in the pixel/feature space to reduce the inter-view redundancy. Meanwhile, Lei et al. (2022) introduces a bi-directional coding framework, called as BCSIC, to jointly compress left and right images simultaneously for exploring the content dependency between the stereo pair. These rudimentary studies demonstrate the potentials of deep neural networks (DNNs) in saving significant bit-rate for MIC.

However, there are several significant shortcomings hampering the deployment and application scope of existing MIC methods. **Firstly**, both the traditional and learning-based approaches demand inter-view prediction at the encoder, *i.e.*, joint encoding, which requires the cameras to communi-

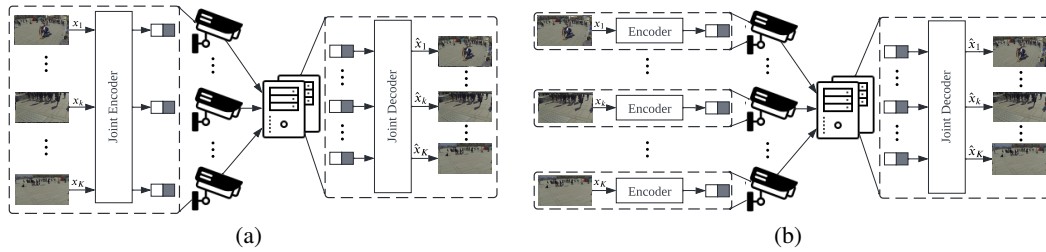

Figure 1: Overview of different multi-view image coding architectures, including (a) a joint encoding architecture and (b) the proposed symmetric distributed coding architecture.

cate with each other or to transmit the data to an intermediate common receiver, thereby consuming a tremendous amount of communication resources and increasing the deployment cost (Gehrig & Dragotti, 2007). This is undesirable in applications relevant to wireless multimedia sensor networks (Akyildiz et al., 2007). An alternative is to deploy special sensors like stereo cameras as the encoder devices to acquire the data, but these devices are generally more expensive than monocular sensors and suffer from limited field of view (FoV) due to the constraints of distance and position between built-in sensors (Li, 2008). **Secondly**, most of the prevailing schemes, except BCSIC, are developed based on disparity correlations defined by the epipolar geometric constraint (Scharstein & Szeliski, 2002), which usually requires to know the internal and external parameters of the camera in advance, such as camera locations, orientations, and camera matrices. Whereas, it is difficult for a distributed camera system without communication to access the prior knowledge of cameras (Devarajan et al., 2008). For example, the specific location information of cameras in autonomous driving is usually not expected to be perceived by other vehicles or infrastructure in order to avoid leaking the location and trajectory of individuals (Xiong et al., 2020). **Finally**, as shown in Table 1 and Figure 4, compared with state-of-the-art (SOTA) learning-based single image codecs (Minnen et al., 2018; Cheng et al., 2020), existing DNN-based MIC methods are not competitive in terms of rate-distortion (RD) performance, which is potentially caused by inefficient inter-view prediction networks.

To address the above challenges, we resort to innovations in the image coding architecture. Particularly, our inspiration comes from the Slepian-Wolf (SW) theorem (Slepian & Wolf, 1973; Wolf, 1973) on distributed source coding (DSC) [1]. The SW theorem illustrates that separate encoding and joint decoding of two or more correlated sources can theoretically achieve the same compression rate as a joint encoding-decoding scheme under lossless compression. It has been extended to the lossy case by Berger (1978) and Tung (1978), which provides the inner and outer bounds of the achievable rate region. Based on these information-theoretic results on DSC, we develop a learning-based distributed multi-view image coding (LDMIC) framework. Specifically, **to avoid collaboration between different cameras**, as shown in Figure 1(b), each view image is mapped to the corresponding quantized latent representation by an individual encoder, while a joint decoder is used to reconstruct the whole image set, which can successfully avoid the communication among cameras or the usage of special sensors. This architectural innovation is theoretically supported by the DSC theory. **Instead of disparity-based correlations**, we design a joint context transfer (JCT) module based on the cross-attention mechanism agnostic to geometry priors to exploit the global content dependencies between different views at the decoder, making our approach applicable to arbitrary multi-camera systems with overlapping FoV. **Finally**, since the separate encoding and joint decoding scheme is implemented by DNNs, the end-to-end RD optimization strategy is leveraged to implicitly help the encoder to learn to remove the partial inter-view redundancy, thus improving the compression performance of the overall system. In summary, our main contributions are as follows:

- To the best of our knowledge, this is the first work to develop a novel deep learning-based *view-symmetric* framework for multi-view image coding. It decouples the inter-view operations at the encoder, which is highly desirable for distributed camera systems.

- We present a joint context transfer module at the decoder to explicitly capture inter-view correlations for generating more informative representations. We also propose an end-to-end encoder-decoder training strategy to implicitly make the latent representations more compact.

---

[1]More details about the theorem and proposition of distributed source coding are provided in Appendix 6.4.

- Extensive experimental results show that our proposed framework is the first distributed codec achieving comparable coding performance to the SOTA joint encoding-decoding schemes, implying the effectiveness of the inter-view cross-attention mechanism compared with the conventional disparity-based prediction. Moreover, our proposed framework outperforms the asymmetric-based coding framework NDIC (Mital et al., 2022b), which demonstrates the advantage of the view-symmetric design over the asymmetric one.

## 2 RELATED WORKS

**Single Image Coding.** In the past decades, various standard image codecs have been developed, including JPEG (Wallace, 1992), JPEG2000 (Skodras et al., 2001), BPG (Bellard, 2014), and VVC intra (Bross et al., 2021). They generally apply three key ideas to reduce redundancy: (i) transform coding, e.g., discrete cosine transform, to decrease the spatial correlation, (ii) quantization of transform coefficients to filter the irrelevancy related to the human visual system, and (iii) entropy coding to lessen the statistical correlation of the coded symbols. Unfortunately, these components are separately optimized, making it hard to achieve optimal coding efficiency.

Recently, end-to-end image compression has engaged increasing interests, which is built upon the transform coding paradigm with nonlinear transform and powerful entropy models for higher compression efficiency. Nonlinear transform is used to produce compact representations, such as generalized divisive normalization (GDN) (Ballé et al., 2015), the self-attention block (Cheng et al., 2020), wavelet-like invertible transform (Ma et al., 2020) and stacks of residual bottleneck blocks (He et al., 2022). To approximate the distribution of latent representations, many advanced entropy models have been proposed. For example, Ballé et al. (2017; 2018) put forward the factorized and hyper prior entropy models for the first time. Then the auto-regressive context model (Minnen et al., 2018) is combined into the hyper prior to effectively reduce the spatial redundancy of images at the expense of high decoding latency. In order to improve the decoding speed, Minnen & Singh (2020) and He et al. (2021) investigate the channel-wise and spatial-wise context versions, respectively. These existing works are considered as important building blocks for our scheme.

**Multi-view Image Coding.** Conventional MIC standards (Vetro et al., 2011; Tech et al., 2015) are derived from key frame compression methods designed for multi-view video codecs. Since these methods are still in the development stage and only support YUV420 format, they are uncompetitive against single image codecs that allow the YUV444 or RGB format. Meanwhile, existing learning-based MIC approaches (Liu et al., 2019; Deng et al., 2021; Wödlinger et al., 2022; Lei et al., 2022) mainly focus on stereo images, and it is difficult to effectively extend them to the general multi-view scenario. Moreover, they can only handle a fixed number of views. In contrast, our framework exerts average pooling to merge the information between multiple views, making it insensitive to the number of viewpoints.

**Distributed Source Coding.** There have been some works developing multi-view compression methods based on DSC. They are typically built on the setting of coding with side information (Zhu et al., 2003; Thirumalai et al., 2007; Chen et al., 2008; Wang et al., 2012), where one view is selected as a reference and compressed independently. For other views, the joint decoder uses the reference as side information to capture the inter-view correlations to reduce the coding rate. Recent learning-based distributed multi-view image compression concentrates on this asymmetric paradigm (Ayzik & Avidan, 2020; Whang et al., 2021; Wang et al., 2022; Mital et al., 2022a;b). Nevertheless, this architecture suffers from high transmission cost for the primary sensor, since it requires a hierarchical relationship between different cameras, leading to the unbalanced coding rates among them (Tosic & Frossard, 2009).

Different from the above works, we consider a more practical symmetric coding pattern illustrated in Figure 1(b), where all cameras are treated as equal status. While traditional symmetric coding schemes (Thirumalai et al., 2008; Gehrig & Dragotti, 2009) utilize disparity-based estimation at the decoder to reduce the transmission cost, we get rid of the disparity compensation prediction and adopt the cross-attention mechanism (Vaswani et al., 2017) to capture the global relevance between different views, which effectively improves the compression performance and broadens the application scope. As far as our knowledge, our study is the first in applying DNNs into symmetric distributed coding and achieving the RD performance comparable to joint encoding-decoding schemes.

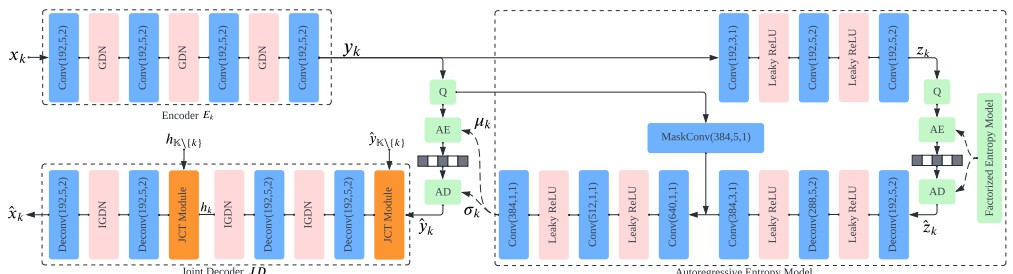

Figure 2: The proposed LDMIC framework with an auto-regressive entropy model, where $\hat{\boldsymbol{y}}_{\mathbb{K}\setminus\{k\}}$ and $\boldsymbol{h}_{\mathbb{K}\setminus\{k\}}$ represent the set of all the view features except for the $k$-th view feature $\hat{\boldsymbol{y}}_k$ and $\boldsymbol{h}_k$, respectively. Convolution/deconvolution parameters are formatted as (the number of output channels, kernel size, stride). Q denotes quantization. AE and AD represent arithmetic encoder and decoder, respectively.

## 3 PROPOSED METHOD

### 3.1 THE OVERALL ARCHITECTURE OF LDMIC

Figure 2 depicts the network architecture of the proposed method. Let $\mathbb{K} = \{1, \cdots, K\}$ denote the image index set. Given a group of multi-view images $\boldsymbol{x}_{\mathbb{K}} = \{\boldsymbol{x}_1, \boldsymbol{x}_2, \cdots, \boldsymbol{x}_K\}$, each image $\boldsymbol{x}_k$ is independently mapped to the corresponding representation $\boldsymbol{y}_k$ by the encoder $E_k$ with shared network parameters. Then $\boldsymbol{y}_k$ is quantized to $\hat{\boldsymbol{y}}_k$. After receiving all the quantized representations $\hat{\boldsymbol{y}}_{\mathbb{K}}$, the joint decoder $JD$ exploits the inter-view correlations among $\hat{\boldsymbol{y}}_{\mathbb{K}}$ to reconstruct the whole image set $\hat{\boldsymbol{x}}_{\mathbb{K}}$. The compression procedure is described as

$$
\begin{aligned}
\boldsymbol{y}_k &= E_k(\boldsymbol{x}_k, \boldsymbol{\phi}), \forall k \in \mathbb{K}, \\
\hat{\boldsymbol{y}}_k &= Q(\boldsymbol{y}_k), \forall k \in \mathbb{K}, \\
\hat{\boldsymbol{x}}_{\mathbb{K}} &= JD(\hat{\boldsymbol{y}}_{\mathbb{K}}; \boldsymbol{\theta}),
\end{aligned}
\tag{1}
$$

where $\boldsymbol{\phi}$ and $\boldsymbol{\theta}$ are optimized parameters of the encoder and decoder. Since the quantizer $Q$ is not differentiable, we apply the mixed quantization approach proposed in Minnen & Singh (2020) during training. Specifically, the latent representation $\boldsymbol{y}_k$ with an additive uniform noise is taken as the input to the entropy model for estimating the bitrate, while the rounded representation with a straight-through-estimation (STE) gradient flows to the joint decoder for reconstruction.

To apply entropy coding to reduce the statistical correlation of the quantized representation $\hat{\boldsymbol{y}}_k$, each element $\hat{y}_{k,i}$ is modelled as a univariate Gaussian random variable with its mean $\mu_{k,i}$ and standard deviation $\sigma_{k,i}$ by introducing a side information $\hat{z}_{k,i}$, where $i$ denotes the position of each element in a vector-valued signal. The probability distribution $p_{\hat{\boldsymbol{y}}_k|\hat{\boldsymbol{z}}_k}$ of $\hat{\boldsymbol{y}}_k$ is expressed as follows:

$$
p_{\hat{\boldsymbol{y}}_k|\hat{\boldsymbol{z}}_k}(\hat{\boldsymbol{y}}_k|\hat{\boldsymbol{z}}_k) \sim \mathcal{N}(\boldsymbol{\mu}_k, \boldsymbol{\sigma}_k^2).
\tag{2}
$$

Meanwhile, a context model is also combined with the entropy model for effectively reducing the spatial redundancy of latent $\hat{\boldsymbol{y}}_k$. The selection of the context model depends on the specific needs of different applications. We choose an auto-regressive model (Minnen et al., 2018) and a checkerboard model (He et al., 2021) for better coding efficiency and faster coding speed, respectively.

### 3.2 JOINT CONTEXT TRANSFER MODULE

Due to the overlap between the cameras' FoV, there exist significant inter-view correlations in the feature space, which inspires us to propose a joint context transfer (JCT) module to exploit this property for generating more informative representations. As shown in Figure 3, the proposed JCT module receives multi-view features $\boldsymbol{f}_{\mathbb{K}}$ as inputs, learns an inter-view context for each view feature, and refines the input features based on the corresponding inter-view contexts. Note that there are $K$ parallel paths in the JCT module. Each path shares the same network parameters and follows a three-step process described below to obtain the refined representations $\boldsymbol{f}_{\mathbb{K}}^*$.

**Feature extraction.** We firstly utilize two residual blocks to extract the representative feature $\boldsymbol{f}_k'$ from the $k$-th view $\boldsymbol{f}_k$. Each residual block, as depicted in Figure 3, is composed of two consecutive convolution layers with Leaky ReLU activation functions.

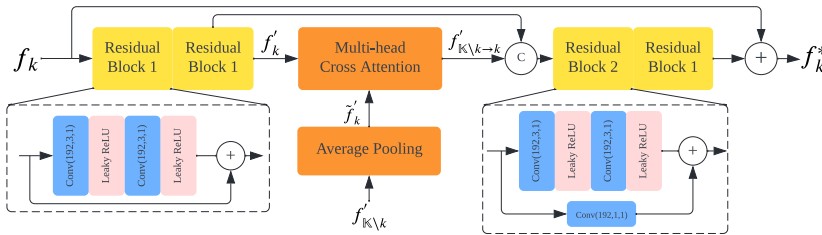

Figure 3: Illustration of the $k$-th path in the proposed joint context transfer module, where $\boldsymbol{f}'_{\mathbb{K}\setminus\{k\}}$ denotes the set of all the view representations except for the current view representation $\boldsymbol{f}'_k$.

**Multi-view fusion.** All the representations $\boldsymbol{f}'_\mathbb{K}$ from the feature extraction module except $\boldsymbol{f}'_k$ are aggregated to a preliminary context $\tilde{\boldsymbol{f}}'_k$ via a simple average pooling over the dimension of the number of the input features:

$$\tilde{\boldsymbol{f}}'_k = \frac{1}{K-1} \sum_{i\in\mathbb{K}\setminus\{k\}} \boldsymbol{f}'_i, \tag{3}$$

where $\mathbb{K}\setminus\{k\} = \{1,\cdots,k-1,k+1,\cdots,K\}$. By this aggregation operation, we achieve fusion between any number of view features. In addition, it is observed that more complex pooling approaches can be developed to further improve the performance.

After getting the aggregated context, we apply a multi-head cross-attention module to exploit the dependency between $\boldsymbol{f}'_k$ and $\tilde{\boldsymbol{f}}'_k$. Since the original attention module incurs high memory and computational cost under a large spatial dimension of input, we adopt the resource-efficient attention in Shen et al. (2021). Specifically, we use a $1 \times 1$ convolution layer and a reshape operation to transform $\boldsymbol{f}'_k \in \mathbb{R}^{H\times W\times d}$ and $\tilde{\boldsymbol{f}}'_k \in \mathbb{R}^{H\times W\times d}$, i.e., query $\boldsymbol{Q}_k = \text{Conv}(\boldsymbol{f}'_k) \in \mathbb{R}^{n\times h\times d_1}$, key $\boldsymbol{K}_k = \text{Conv}(\tilde{\boldsymbol{f}}'_k) \in \mathbb{R}^{n\times h\times d_1}$ and value $\boldsymbol{V}_k = \text{Conv}(\tilde{\boldsymbol{f}}'_k) \in \mathbb{R}^{n\times h\times d_2}$ , where $n = H \times W$ and $h$ denotes the number of heads. The notations $d$, $d_1$ and $d_2$ are the channel dimensions of input, key (query) and value in a head, respectively. Then the multi-head cross-attention module is applied as:

$$\boldsymbol{A}_{k,i} = \sigma_{row}(\boldsymbol{Q}_{k,i})(\sigma_{col}(\boldsymbol{K}_{k,i})^\mathsf{T}\boldsymbol{V}_{k,i}), \forall i=1,\cdots,h$$
$$\boldsymbol{f}'_{\mathbb{K}\setminus\{k\}\to k} = \text{Conv}(\boldsymbol{A}_{k,1} \oplus \cdots \oplus \boldsymbol{A}_{k,h}), \tag{4}$$

where $\sigma_{row}$ ($\sigma_{col}$) denotes applying the softmax function along each row (column) of the matrix, and $\oplus$ is the channel-wise concatenation. The context $\boldsymbol{f}'_{\mathbb{K}\setminus\{k\}\to k}$ relevant to the $k$-th view feature is extracted and will be injected into the current feature in the next step.

**Refinement.** Based on the learned inter-view context $\boldsymbol{f}'_{\mathbb{K}\setminus\{k\}\to k}$, the input feature $\boldsymbol{f}_k$ is refined to a more informative feature $\boldsymbol{f}^*_k$:

$$\boldsymbol{f}^*_k = \boldsymbol{f}_k + F(\boldsymbol{f}'_{\mathbb{K}\setminus\{k\}\to k} \oplus \boldsymbol{f}'_k), \tag{5}$$

where $F(\cdot)$ consists of two consecutive residual blocks. As shown in Figure 2, the JCT module is placed before the first and third deconvolution layers to connect the different-view decoding stream for feature aggregation and transformation.

### 3.3 TRAINING

The target of LDMIC is to optimize the trade-off between the number of encoded bits and the reconstruction quality. Therefore, a training loss composed of two metrics is used:

$$L = \lambda D + R = \lambda \sum_{k=1}^{K} d(\boldsymbol{x}_k, \hat{\boldsymbol{x}}_k) + \sum_{k=1}^{K} \big(R(\hat{\boldsymbol{y}}_k) + R(\hat{\boldsymbol{z}}_k)\big) \tag{6}$$

where $d(\boldsymbol{x}_k, \hat{\boldsymbol{x}}_k)$ is the distortion between $\boldsymbol{x}_k$ and $\hat{\boldsymbol{x}}_k$ under a given metric, such as mean squared error (MSE) $R(\hat{\boldsymbol{y}}_k)$ and $R(\hat{\boldsymbol{z}}_k)$ represent the estimated compression rates of the latent representation $\hat{\boldsymbol{y}}_k$ and the corresponding hyper representation $\hat{\boldsymbol{z}}_k$, respectively. $\lambda$ is a hyper parameter that controls the trade-off between the bit rate cost $R$ and distortion $D$.

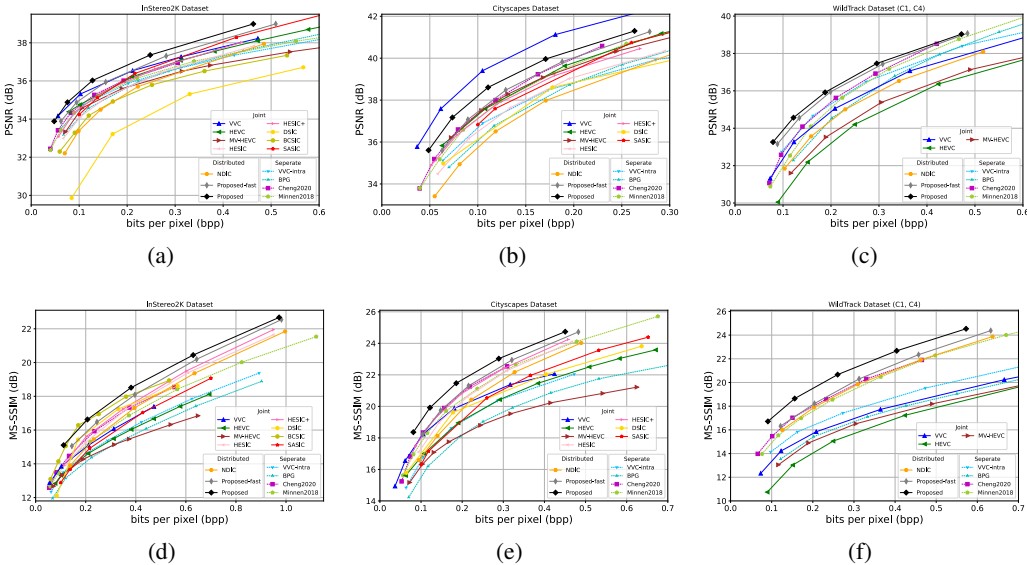

Figure 4: Rate-distortion curves of our proposed methods compared against various competitive baselines.

# 4 EXPERIMENTS

## 4.1 EXPERIMENTAL SETUP

**Datasets.** To compare with the recently developed learning-based stereo image compression methods, two common stereo image datasets, *i.e.*, Instereo2K (Bao et al., 2020) and Cityscapes (Cordts et al., 2016), are chosen to evaluate the coding efficiency of the proposed framework. Apart from testing stereo image datasets related to 3D scenes, we also select a pedestrian surveillance dataset, i.e., WildTrack (Chavdarova et al., 2018), acquired by seven random placed cameras with overlapping FoV, which is to demonstrate the potentials of our proposed framework in distributed camera systems without epipolar geometry relationship between images. More details about datasets are provided in Appendix 6.5.

**Benchmarks.** The competing baselines can be split into three categories: (1) *Separate* model independently compresses each image, whose typical SOTA representatives are BPG (Bellard, 2014), VVC-intra (Bross et al., 2021), Minnen et al. (2018) and Cheng et al. (2020). For BPG and VVC-intra, we disable chroma subsampling. (2) *Joint* model has access to a set of multi-view images and explicitly utilizes the inter-view redundancy to achieve a high compression ratio. According to performance comparisons in Wödlinger et al. (2022), conventional video standards can be applied in the MIC, where each set of multi-view images is compressed as a multi-frame video sequence by using both HEVC (Sullivan et al., 2012) and VVC (Bross et al., 2021) with *lowdelay_P* configuration as well as YUV444 input format. We also test MV-HEVC (Tech et al., 2015) with the multi-view intra mode. Apart from that, we report the results of several recent DNN-based stereo image codecs on the InStereo2K and Cityscapes datasets, including DSIC (Liu et al., 2019), two variants of HESIC (Deng et al., 2021), BCSIC (Lei et al., 2022), and SASIC (Wödlinger et al., 2022). (3) *Distributed* model only uses the joint decoder to implicitly reduce the inter-view dependency. We compare our method with NDIC based on asymmetric DSC (Mital et al., 2022b) to demonstrate the superiority of symmetric DSC. More details on baseline settings are given in Appendix 6.5.

**Metrics.** The distortion between the reconstructed and original images is measured by peak signal-to-noise ratio (PSNR) and multi-scale structural similarity index (MS-SSIM) (Wang et al., 2003). Besides assessing RD curves, we compute the Bjøntegaard Delta bitrate (BDBR) (Bjøntegaard, 2001) results to represent the average bitrate savings at the same distortion level.

**Implementation Details.** We train our models with five different $\lambda$ values, where $\lambda = 256, 512, 1024, 2048, 4096$ (8, 16, 32, 64, 128) under MSE (MS-SSIM). For MSE optimized models, they are trained from scratch for 400 epochs on InStereo2K/Cityscapes and 700 epochs on

Table 1: Comparison of BDBR cost relative to BPG on different datasets, with the best results in **Bold** and second-best ones in underlined.

| Categories | Methods | InStereo2K | | Cityscapes | | WildTrack (C1, C4) | |
|---|---|---|---|---|---|---|---|
| | | PSNR | MS-SSIM | PSNR | MS-SSIM | PSNR | MS-SSIM |
| Separate | Minnen2018 | -7.44% | -34.37% | -21.58% | -46.74% | -10.40% | -47.73% |
| | Cheng2020 | -19.71% | -41.95% | -27.86% | -49.63% | -19.23% | -52.54% |
| | VVC-intra | -3.23% | -17.38% | -4.14% | -22.12% | -10.76% | -24.84% |
| Joint | VVC | -30.54% | -33.80% | **-52.85%** | -46.35% | -4.18% | -9.31% |
| | HEVC | -15.54% | -14.70% | -23.40% | -24.48% | 39.04% | 23.27% |
| | MV-HEVC | 2.83% | -4.75% | -18.57% | -0.17% | 33.88% | 9.35% |
| | HESIC | 0.47% | -39.55% | -7.92% | -45.14% | - | - |
| | HESIC+ | -15.06% | -43.56% | -21.70% | -51.33% | - | - |
| | DSIC | 107.88% | -40.04% | -1.88% | -38.26% | - | - |
| | BCSIC | 23.80% | -56.11% | - | - | - | - |
| | SASIC | -19.83% | -23.04% | -20.39% | -30.10% | - | - |
| Distributed | NDIC | 13.98% | -34.38% | 7.36% | -38.07% | 3.94% | -51.08% |
| | Proposed-fast | -29.68% | -49.89% | -28.30% | -53.61% | -26.69% | -55.08% |
| | Proposed | **-41.69%** | **-59.20%** | -40.14% | **-62.12%** | **-31.21%** | **-67.77%** |

Table 2: Complexity of learning-based image codecs evaluated on a pair of stereo images with the resolution as 832×1024 in the InStereo2K dataset, where the encoding latency of DSC-based schemes is determined by the maximum time for independent encoding of each image.

| Methods | Encoder | | | Decoder | | |
|---|---|---|---|---|---|---|
| | FLOPs | Params | Time | FLOPs | Params | Time |
| DSIC | 2415.29G | 79.26M | 25.97s | 3378.65G | 75.78M | 26.45s |
| HESIC | 285.3G | 32.08M | 3.23s | 1197.22G | 29.55M | 16.15s |
| HESIC+ | 205.71G | 17.02M | 16.79s | 1122.87G | 15.28M | 49.96s |
| SASIC | 531.42G | 3.58M | 10.66s | 2532.87G | 4.48M | 34.45s |
| NDIC | 163.93G×2 | 7.25M×2 | 3.19s | 1245.89G | 25.04M | **9.93s** |
| Proposed-fast | 194.15G×2 | 11.24M×2 | **2.37s** | 1851.96G | 15.24M | 11.48s |
| Proposed | 187.39G×2 | 11.24M×2 | 9.44s | 1838.42G | 15.24M | 47.87s |

WildTrack by using Adam optimizer (Kingma & Ba, 2014), in which the batch size is taken as 8. The learning rate is initially set as $10^{-4}$ and decreased by a factor of 2 every 100 epochs until it reaches 400 epochs. As for MS-SSIM optimized models, we fine-tune the MSE optimized networks for 300 (400) epochs with the initial learning as $5 \times 10^{-5}$ on stereo (multi-camera) image dataset. During training, each image is randomly flipped and cropped to the size of $256 \times 256$ for data augmentation. The whole framework is implemented by CompressAI (Bégaint et al., 2020) and trained on a machine with NVIDIA RTX 3090 GPU.

## 4.2 Experimental Results

**Coding performance.** Figure 4 presents the RD curves of all compared methods and Table 1 gives the corresponding BDBR results of each codec relative to BPG. For InStereo2K and Cityscapes, the proposed method outperforms most of these compression baselines in both PSNR and MS-SSIM, which implies that relying only on joint decoding can effectively reduce the inter-view redundancy between different views. For example, when compared with Cheng2020 (SASIC), our method and the fast variant reduce around 21.98% (21.86%) and 9.97% (9.85%) bits in terms of PSNR, respectively. Since stereo images contain plenty of homogeneous regions suited for traditional coding, VVC achieves up to 30.54% and 52.85% compression efficiency when measured by PSNR, but notice that it requires joint encoding. On the InStereo2K dataset, VVC underperforms our method by a margin with about 0.44dB coding gains in PSNR due to a larger variation in image content. In addition, our proposed framework attains better reconstruction quality measured by MS-SSIM at the same bitrate when compared with VVC.

As seen from Figure 4(c) and 4(f), we select the images acquired by two cameras, C1 and C4, on the WildTrack dataset to evaluate the compression performance of different methods without

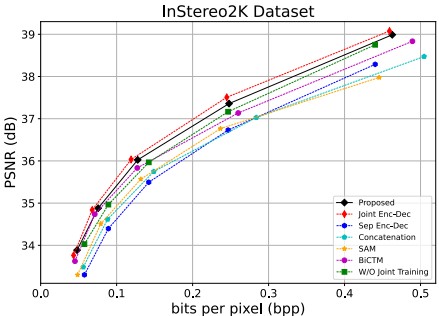

Figure 5: Ablation study. *Joint Enc-Dec* and *Sep Enc-Dec* denote inserting and removing the JCT module at the encoder and decoder, respectively. *Concatenation*, *SAM* and *BiCTM* represent different inter-view operations to replace the proposed JCT module at the decoder. *W/O Joint Training* is to fix the pretrained encoder including the entropy model and only train the joint decoder.

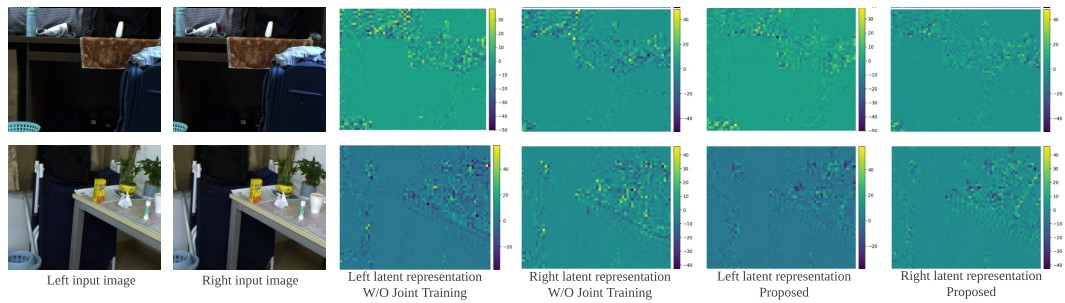

| Left input image | Right input image | Left latent representation W/O Joint Training | Right latent representation W/O Joint Training | Left latent representation Proposed | Right latent representation Proposed |

Figure 6: Visual examples from the InStereo2K dataset, where we assemble all channels of the latent representation $Q(\boldsymbol{y}_k - \boldsymbol{\mu}_k)$ to display the feature map.

using additional information from other cameras. It is observed that the traditional video codecs perform worse than the corresponding intra-frame ones due to lots of heterogeneous overlapping regions, which makes it difficult for standard video codecs to effectively capture the inter-view redundancy by using compensation-based predictions. However, our proposed framework relies on the cross-attention mechanism to exploit the correlations of different views from the perspective of global receptive fields, thereby providing up to 31.21% and 67.77% bitrate saving in PSNR and MS-SSIM, respectively. The remarkable results demonstrate that the proposed LDMIC framework is a promising solution to meet the compression needs of distributed camera systems. The RD curves on the multi-camera case have similar trends with that on the two-camera one, which are provided in Appendix 6.1.

Moreover, compared with asymmetric DSC-based NDIC, the proposed method saves 55.67%, 47.5% and 35.15% bits in PSNR on three datasets (InStereo2K, Cityscapes, WildTrack). For the proposed-fast variant with the checkerboard entropy model, the improvements are also adequate, *i.e.*, 43.66%, 35.66% and 30.63%. This set of results indicate that the usage of bi-directional information based on symmetric DSC can better exploit the inter-view correlations to bring higher coding gains. Additionally, our methods have better compression efficiency in MS-SSIM than in PSNR, which is partly caused by exploiting the inter-view correlations in the feature space rather than pixel space at the decoder. Thus, the network tends to focus on structure information instead of pixel information.

**Computational complexity.** Table 2 shows the computational complexity of seven image codecs running on an Intel Xeon Gold 6230R processor with base frequency 2.10GHz and a single CPU core, including the number of FLOPs, the model parameters and the coding latency. Different from the joint models, our methods designed on DSC decouples the inter-view operations at the encoder, which allows image-level parallel processing. Therefore, the proposed-fast variant enjoys about $1.36 \sim 10.95$ and $1.41 \sim 4.35$ times encoding and decoding speedup against the learned joint schemes (*i.e.*, DSIC, HESIC, HESIC+, SASIC). Even if the auto-regressive entropy model is used, the encoding of our method is still faster than that of both DSIC and SASIC based on hyper

Table 3: Bitrate savings for two-view images with cameras C1 and C2 as the number of viewpoints increases on the WildTrack dataset, where the case of $K = 2$ is set as the anchor.

| Number of cameras | $K = 2$ | $K = 3$ | $K = 4$ | $K = 5$ | $K = 6$ | $K = 7$ |
|---|---|---|---|---|---|---|
| Bitrate saving (%) | 0 | 0.0053 | 0.0801 | 1.0919 | 1.4161 | 1.5004 |

prior. Moreover, our proposed fast variant with better coding efficiency achieves similar coding time with another DSC-based method NDIC, which demonstrates the superiority of symmetric DSC in coding speed and compression efficiency. For more details on comparison between our methods and traditional codecs, please refer to Appendix 6.2.

### 4.3 ABLATION STUDY

**Inter-view Fusion.** To verify the contribution of the JCT module for fusing inter-view information, a set of ablation experiments are conducted on the InStereo2K dataset with RD curves shown in Figure 5. Specifically, we allow (forbid) both the encoder and the decoder to access the inter-view context, which provides an upper (lower) bound on the performance of the proposed method and is denoted by *Joint (Sep) Enc-Dec*. In this case, the PSNR with (without) the JCT module at the encoder (decoder) improves (drops) by about 0.16dB (0.73dB) at the same bpp level. We further report the compression results when the JCT module is directly replaced by other inter-view fusion operations such as concatenation in Mital et al. (2022b), stereo attention module (SAM) in Wödlinger et al. (2022) and bi-directional contextual transform module (Bi-CTM) in Lei et al. (2022). These operations lead to an increase of the bitrate by 32.73%, 27.99%, 10.11% compared with our method. The experimental results indicate that our proposed JCT module have powerful capability in capturing inter-view correlations and generating more informative representations.

**Joint Training Strategy.** In this paper, we exploit the benefit of joint training to implicitly help the encoder to learn removing the partial inter-view redundancy. Thus, the latent representation is expected to be more compact. To investigate its effect, we perform a experiment by only training the joint decoder with the fixed pre-trained encoder and entropy model. As shown in Figure 5, our approach outperforms the *W/O Joint Training* method by 0.225 dB. In Figure 6, we provide further visual comparisons. It is noted that the latent feature maps with joint training strategy contain more elements with low magnitudes, which requires much fewer bits for encoding.

**Number of views.** Table 3 shows the impact of different numbers of views on coding efficiency. We compare the bitrate of cameras C1 and C2 when incorporating different numbers of views during decoding. The bitrate saving increases gradually as more information is received from different cameras. Due to only using a simple average pooling to merge multi-view information to the inter-view context, we get a marginal coding gains when incorporating more views. It is possible to further improve the compression gains of our framework by using more complex aggregation approaches.

## 5 DISCUSSION

In this paper, we presented a novel end-to-end distributed multi-view image coding framework nicknamed LDMIC. Our proposal inherits the advantages of traditional distributed compression in image-level parallelization, which is desirable for distributed camera systems. Meanwhile, leveraging the insensitivity of the cross-attention mechanism to epiploar geometric relations, we develop a joint context transfer module to account for global correlations between images from different viewpoints. Experimental results demonstrate the competence of LDMIC in achieving higher coding gains than existing learning-based joint and separate encoding-decoding schemes. Moreover, compared with learned joint models, the LDMIC fast variant enjoys a much lower coding complexity with on-par compression performance. To the best of our knowledge, this is the first successful attempt of the distributed coding architecture to fight against the performance of the joint coding paradigm under the lossy compression case.

Based on the proposed framework, there are two clear directions to be explored in the future. On one hand, as mentioned in Section 4.3, it is interesting to investigate how to more effectively incorporate different view information to generate a better inter-view context. One the other hand, it is worth exploring how to extend the framework to multi-view video compression.

ACKNOWLEDGMENTS

This work was supported by the NSFC/RGC Collaborative Research Scheme (Project No. CRS_HKUST603/22).

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

# 6 APPENDIX

## 6.1 RD CURVES ON MULTI-CAMERA WILDTRACK DATASET

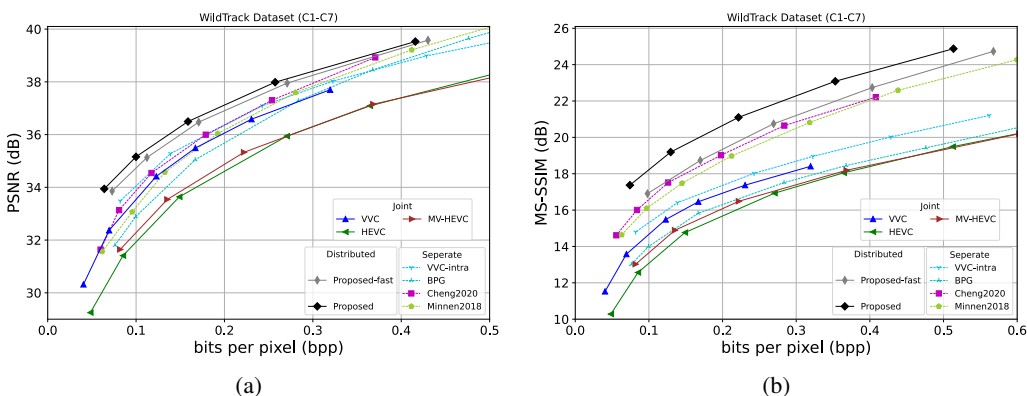

Figure 7: Comparison of compression efficiency on WildTrack dataset with seven views.

## 6.2 CODING COMPLEXITY

Figure 8 reports the coding latency of different codecs on an Intel Xeon Gold 6230R processor with a single CPU core. For the proposed methods, we also evaluate the inference latency on a workstation with an NVIDIA RTX 3090 GPU. On the CPU platform, the proposed methods achieve tremendous encoding speedup improvement against VVC, which benefits from parallel processing all images in the DSC architecture. Because of the auto-regressive model and computation resources constraint, the decoder has a large latency. The proposed framework targets for applications related to distributed camera systems, such as video surveillance and multi-view image acquisition. These applications require a low-power encoder, while the receiver has powerful computation resources to support decoding procedure. As depicted in Figure 8(b), the proposed-fast variant on the GPU platform consumes less decoding time and outperforms HEVC with 14.14% bitrate saving measured by PSNR. When compared with VVC, the fast variant with only 0.86% increase in bits reduces about 50% decoding time. The results demonstrate that the decoding latency of proposed methods with GPU support can meet the basic needs.

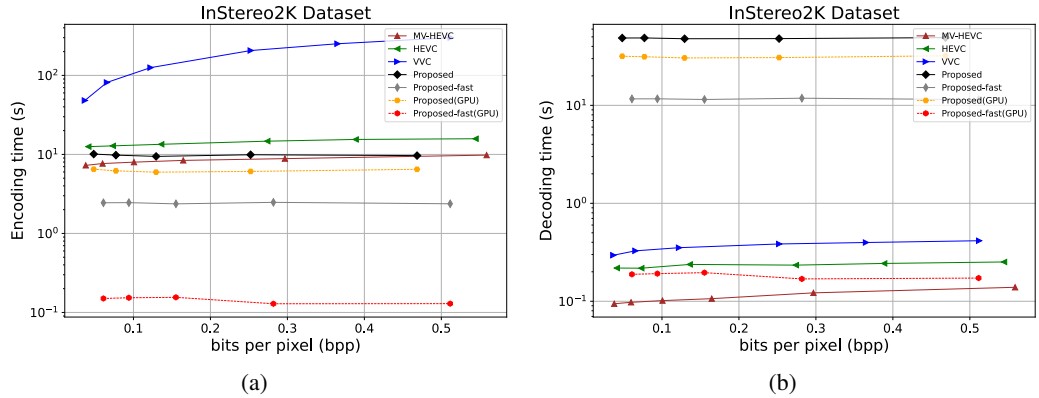

Figure 8: Encoding and decoding time of proposed methods and traditional codecs on InStereo2K dataset.

## 6.3 VISUALIZATIONS

In Figure 9, we present several examples to vividly compare the quantitative results among Cheng2020, VVC, NDIC and the proposed method. It is observed that our proposed method effectively restores the image details and maintains higher reconstruction quality while consuming the

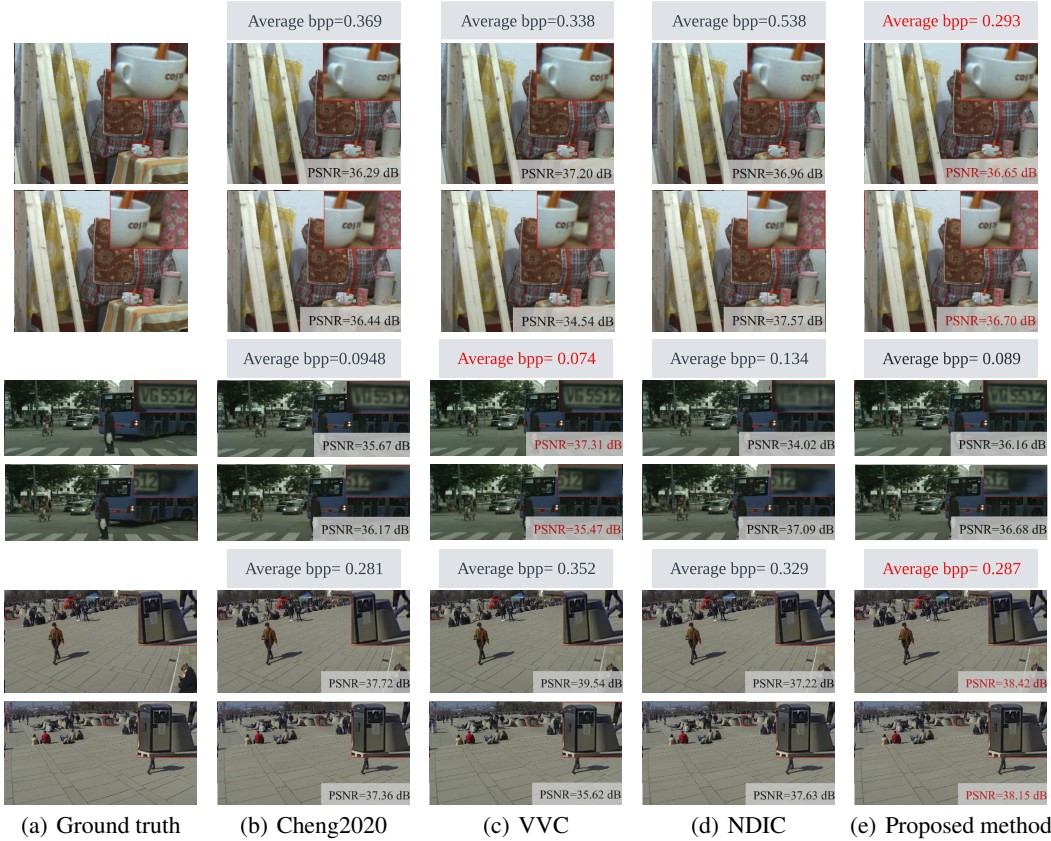

|  (a) Ground truth | (b) Cheng2020 | (c) VVC | (d) NDIC | (e) Proposed method |

Figure 9: A subjective comparisons on the InStereo2K, Cityscapes and WildTrack datasets, where the best results are outlined in red color.

lower bits on the InStereo2K and WildTrack datasets. Similar as the results in Figure 4(b), VVC achieves the best coding gain on the Cityscapes dataset.

## 6.4 FOUNDATIONS OF SYMMETRIC DISTRIBUTED SOURCE CODING

The formal statements of Slepian-Wolf theorem (Slepian & Wolf, 1973; Wolf, 1973) and Berger–Tung proposition (Berger, 1978; Tung, 1978; Servetto, 2006) are as follows.

**Theorem 1 (Slepian-Wolf)** *Let $X_1$ and $X_2$ be two statistically dependent i.i.d. discrete sources. The achievable rate region of independently encoding $X_1$ and $X_2$ with joint decoding under lossless compression is specified by:*

$$R_1 \geq H(X_1|X_2), R_2 \geq H(X_2|X_1), R_1 + R_2 \geq H(X_1, X_2),$$

*where $R_1$ and $R_2$ are the rates for representing $X_1$ and $X_2$, respectively.*

**Proposition 1 (Berger–Tung Bound)** *Let $U_1$ and $U_2$ be auxiliary variables such that there exist decoding functions $\hat{X}_1 = f_1(U_1, U_2)$ and $\hat{X}_2 = f_2(U_1, U_2)$. Given the distortion constraints $E[d(X_j, \hat{X}_j)] \leq D_j, j = 1, 2$, the rates $(R_1, R_2)$ follows the rate region $R_1 \geq I(X_1, X_2; U_1|U_2), R_2 \geq I(X_1, X_2; U_2|U_1), R_1 + R_2 \geq I(X_1, X_2; U_1, U_2)$, for some joint distribution $p(x_1, x_2, u_1, u_2)$.*

· **Inner Bound**: *when $p(x_1, x_2, u_1, u_2)$ satisfies a Markov chain $U_1 - X_1 - X_2 - U_2$, all rates $(R_1, R_2)$ are achievable.*

· **Outer Bound**: *when $p(x_1, x_2, u_1, u_2)$ satisfies two Markov chain $U_1 - X_1 - X_2$ and $X_1 - X_2 - U_2$, those rate points outside the union composed of the set of rates defined for each such $p(x_1, x_2, u_1, u_2)$ are not available.*

The Slepian-Wolf theorem and Berger-Tung bound proposition investigate the lossless and lossy compression of two correlated sources with separate encoders and a joint decoder, respectively. Although until now the compression limit of symmetric coding in the lossy case is still open, these theoretical results indicate that it is possible to compress two statistically dependent signals in a distributed way while approaching the compression performance of joint encoding and decoding.

## 6.5 Experimental Details

**Dataset.** We take two public stereo image datasets, InStereo2K (Bao et al., 2020) and Cityscapes (Cordts et al., 2016), and a multi-camera dataset, WildTrack (Chavdarova et al., 2018), for evaluation. The InStereo2K dataset involves 2060 image pairs for close views and indoor scenes, where 2010 and 50 pairs are selected as the training and testing data, respectively. The Cityscapes dataset is comprised of 5000 image pairs for far views and outdoor scenes, which is categorized into 2975 training, 500 validation and 1525 testing pairs. For the WildTrack dataset, we use *FFMPEG* to extract the images from seven HD 1080 videos at one frame per second. We choose the first 2000 images and the remaining 51 images in each view for training and testing. During evaluation, we minimally crop each image on the InStereo2K dataset so that both height and width are multiples of 64. As for the Cityscapes dataset, we follow the same cropping operations in Wödlinger et al. (2022) to remove rectification artefacts and ego-vehicle, where 64, 256 and 128 pixels from the top, bottom, and sides in each image are cut off.

**Traditional baseline codecs.** We use the evaluation script from CompressAI [2] to obtain the results of conventional codecs. Specifically, instead of using the default x265 encoder in BPG, we adopt the slower but efficient JCTVC encoder option to achieve the higher compression performance. For HEVC and MV-HEVC, the results on the stereo image datasets come from Wödlinger et al. (2022). We use HM-16.25 [3] and HTM-16.3 [4] softwares to evaluate the coding efficiency of HEVC and MV-HEVC on the WildTrack dataset, respectively. In addition, we run VTM-17.0 [5] to test VVC-intra and VVC.

**Learning-based benchmarks.** In DNN-based stereo image codecs, we retest HESIC and HESIC+ including the post-processing network by using their open source codes [6], because they previously reported the wrong results in their paper. The results of DSIC, BCSIC and SASIC are quoted from their corresponding papers. BCSIC did not report the rate-distortion points on the Cityscapes dataset. For distributed models, NDIC is composed of two different models, where one is a single image codec used in Ballé et al. (2018), another consists of separate encoder and joint decoder with side information proposed in Mital et al. (2022b).

**Architecture details.** Details about the network layers in our framework with auto-regressive entropy model are outlined in Figure 2 and 3. For the multi-head attention of the JCT module, we set the number of head as 2. The channel dimensions of the key and value are taken as one eighth and a quarter of input channels (*i.e.*, 48 and 24), respectively. In order to achieve faster coding speed, the proposed fast variant replaces the serial auto-regressive entropy model with the parallelization-friendly checkerboard entropy model in He et al. (2021), which has the same network architecture as Figure 2 except that the masked convolution layer uses a checkerboard mask.

**Ablation study details.** Based on the proposed method, we insert two JCT modules after the second and fourth convolution layers at the encoder to implement the *Joint Enc-Dec*, thereby allowing both the encoder and the decoder to access the inter-view context. For the *Sep Enc-Dec*, the JCT modules at the decoder are removed, making it equivalent to single image compression. These models are trained based on the InStereo2K dataset by using the same training scheme as LDMIC (See *Implementation Details* in Section 4.1). For the *W/O Joint Training* case, we fix the pre-trained encoder and entropy model on the *Sep Enc-Dec*, and only train the joint decoder on the InStereo2K dataset, which follows the same training procedure as in our proposed method.

---

[2]https://github.com/InterDigitalInc/CompressAI/tree/master/compressai/utils

[3]https://vcgit.hhi.fraunhofer.de/jvet/HM/-/tags

[4]https://vcgit.hhi.fraunhofer.de/jvet/HTM/-/tags

[5]https://vcgit.hhi.fraunhofer.de/jvet/VVCSoftware_VTM/-/tags

[6]https://github.com/ywz978020607/HESIC

