# OpenReview forum: "LDMIC: Learning-based Distributed Multi-view Image Coding"
_ICLR.cc/2023/Conference — ICLR 2023 poster_

### Official Review · Reviewer_WX5p · 2022-10-17

**Confidence:** 3
**Clarity, Quality, Novelty And Reproducibility:** This paper is well-written and the or…
**Correctness:** 3
**Technical Novelty And Significance:** 2
**Empirical Novelty And Significance:** 2
**Recommendation:** 6

**Strength And Weaknesses:**

Strength
- This paper introduces a distributed multi-view image coding framework that decouples the inter-view operations at the encoder, which is a good property in a distributed camera system.
- This paper proposes a joint context transfer module at the decoder to capture inter-view correlations for image reconstruction.
- Extensive experimental results show that the proposed method outperforms existing single and multi-view image codecs in PSNR.

Weaknesses
- The authors claim that the proposed joint context transfer module "can serve as a plug-and-play component to enhance the performance of learning-based MIC methods" in the last paragraph of Page 2. However, no supporting experiments are shown
- It appears to me that the technical part of the proposed method is straightforward and not very novel.
- It is unclear what is the fast variant of the proposed method. Is it the one using the checkerboard model? Please clarify.


**Summary Of The Paper:**

This paper introduces a learning-based distributed multi-view image coding (LDMIC) method. LDMIC consists of independent encoders and a decoder equipped with a cross-attention mechanism based joint context transfer module. The proposed method does not need synchronization between cameras and is insensitive to the epipolar geometry relations between images. Experimental results show that LDMIC outperforms existing multi-view image coding methods in saving bitrate.


**Summary Of The Review:**

This paper introduces a simple yet effective method for distributed multi-view image coding, and experiments demonstrate the effectiveness of the method. I would like to give a weak accept rating. However, one of my concern is that the problem tackled by this paper (multi-view image coding) seems to be very specific and only attract a limited amount of audiences in ICLR.

---

> ### Author Response · Authors · 2022-11-08
> **Response to reviewer WX5p**
>
> **Comment 4: However, one of my concern is that the problem tackled by this paper (multi-view image coding) seems to be very specific and only attract a limited amount of audiences in ICLR.**
>
> In the fields of autonomous driving, virtual reality and video surveillance, multi-view image coding has become one of the most critical techniques, which recently attracts increasing attention from both academic and industrial communities. Besides, our proposed method is a successful attempt for applying distributed source coding to the multi-view image compression. Considering distributed source coding has important applications in mono-view and multi-view video coding, our proposed method can enlighten how to apply distributed source coding to design high-efficient and low-complexity image and video coding for uplink-based applications. Thus, we believe our work has a broad impact on the area of generative models.

---

> ### Author Response · Authors · 2022-11-08
> **Response to reviewer WX5p**
>
> Many thanks for your valuable comments and the recognition of our contributions. We hope the following responses could address your concerns.
>
> **Comment 1: The authors claim that the proposed joint context transfer module "can serve as a plug-and-play component to enhance the performance of learning-based MIC methods" in the last paragraph of Page 2. However, no supporting experiments are shown.**
>
> We apology for the confusion caused by the inaccurate description. Our proposed joint context transfer (JCT) module is suitable for the learning-based MIC methods supporting bidirectional decoding. Specifically, we cannot insert our proposed modules into DSIC, HESIC and NDIC frameworks, because these methods adopt a unidirectional decoding mechanism, which requires the previous decoded image as input to the joint decoder.
>
> Due to the lack of architecture details in BCSIC, we only report the BD-Bitrate and BD-PSNR results of the case where the stereo attention module (SAM) in SASIC is replaced with the proposed JCT module. The BD-Bitrate represents the average bitrate savings at the same distortion level. The BD-PSNR represents the gain (dB) when compared with the baseline algorithm at the same bit rate. As shown in the below table, the proposed JCT operation improves by about 0.11dB at the same bpp level and brings a decrease of the bitrate by 3.78\% compared with the original SAM in SASIC on the InStereo2K dataset, which demonstrates that our proposed JCT module can enhance the compression performance of learning-based MIC methods that supports bidirectional decoding.
>
> | Method      | BD-Bitrate | BD-PSNR |
> | ----------- | ---------- | ------- |
> | SASIC (SAM) | 0\%        | 0dB     |
> | SASIC (JCT) | -3.78\%    | 0.11dB  |
>
> **Comment 2: It appears to me that the technical part of the proposed method is straightforward and not very novel.**
>
> With regard to the novelty and technical contribution of our work, the statements in the previous manuscript may not be clear. Thus, we clarify this point here and revise the manuscript accordingly.
>
> - To the best of our knowledge, we are the first work to develop a novel deep learning-based _view-symmetric_ framework for multi-view image coding. It decouples the inter-view operations at the encoder, which is highly desirable for distributed camera systems.
>
> - We present a joint context transfer module at the decoder to explicitly capture inter-view correlations for generating more informative representations and propose an end-to-end encoder-decoder training strategy to implicitly make the latent representation more compact.
>
> - Extensive experimental results show that our proposed framework is the first distributed codec to achieve a coding performance comparable to the state-of-the-art joint encoding-decoding schemes. For example, when compared with SASIC, our method and the fast variant reduce around 21.86\% and 9.85\% bits in terms of PSNR, respectively. It verifies the effectiveness of inter-view cross-attention mechanism compared to the conventional disparity-based prediction.
>   Besides, compared with the _asymmetric-based_ coding framework NDIC, our proposed method achieves higher coding efficiency. This demonstrates the advantage of the view-symmetric design.
>
> **Comment 3: It is unclear what is the fast variant of the proposed method. Is it the one using the checkerboard model? Please clarify.**
>
> The fast variant of the proposed method uses the parallelization-friendly checkerboard entropy model in [1] instead of the serial auto-regressive entropy model. The checkerboard entropy model allows parallelization encoding and decoding, which can achieve faster inference speed. Specifically, the latent representation $\boldsymbol{\hat{y}}$ is split to anchors $\boldsymbol{\hat{y}}\_{1}$ and non-anchors $\boldsymbol{\hat{y}}\_{2}$ in spatial dimension. When encoding/decoding the latent representation $\boldsymbol{\hat{y}}$, the entropy parameters of anchors $\boldsymbol{\hat{y}}\_{1}$ are first calculated by only using the side information $\boldsymbol{\hat{z}}$. After compressing/decompressing $\boldsymbol{\hat{y}}\_{1}$, the entropy parameters of non-anchors $\boldsymbol{\hat{y}}\_{2}$ are calculated from the side information $\boldsymbol{\hat{z}}$ and context $\boldsymbol{\hat{y}}\_{1}$. We have added more details about the fast variant of the proposed method in the revised paper.
>
> [1] Dailan He, Yaoyan Zheng, Baocheng Sun, Yan Wang, and Hongwei Qin. Checkerboard context model for efficient learned image compression. In Proceedings of the IEEE/CVF Conference on Computer Vision and Pattern Recognition, pp. 14771–14780, 2021.

---

> ### Author Response · Authors · 2022-11-16
> **Response to reviewer WX5p**
>
> Many thanks for your valuable comments. The manuscript has been revised and the main changes are highlighted in blue. If you have any questions, please do not hesitate to mention it to us. We are more than happy to address your concerns.

---

### Official Review · Reviewer_8AGJ · 2022-10-20

**Confidence:** 3
**Correctness:** 3
**Technical Novelty And Significance:** 3
**Empirical Novelty And Significance:** 3
**Recommendation:** 6

**Clarity, Quality, Novelty And Reproducibility:**

Clarity, Quality, and Reproducibility: The paper is clear and easy to follow. The proposed method was presented to the details. However, I have several suggestions to improve the clarity and reproducibility.

- The meaning of "the fast variant" of the proposed method is not clear enough. I guess this refers to the one where the auto-regressive model is replaced by the checkerboard model in the context modeling for entropy coding.

- The competing baselines are categorized in the text. It would be better if this categorization was clearly presented in Fig. 4 and Table 1 as well.

- More implementation details for the ablation study is helpful for readers. For example, I cannot imagine how Joint Enc-Dec was implemented. Moreover, the authors did not mention how "the pre-trained encoder" was obtained (what were the condition and dataset for the pre-training) for the w/o joint training case.

Novelty: I think this work has sufficient novelty for presentation at ICLR. Although there are some previous works on distributed coding of multi-view images, this is the first work to develop a view-symmetric framework with deep learning and achieve the coding performance comparable to the SOTA joint encoding-decoding schemes.

**Details Of Ethics Concerns:**

No specific concerns.

**Strength And Weaknesses:**

Strengths

- The paper is well written and easy to follow. The story (background) behind this work is well composed in a succinct manner. The proposed method is clearly motivated and well explained to the details.

- Although there are some previous works on distributed coding of multi-view images, this is the first work to develop a view-symmetric framework with deep learning and achieve the coding performance comparable to the SOTA joint encoding-decoding schemes.

- The experimental evaluation is extensive with convincing results. Especially, the proposed method clearly outperforms NDIC (the method closest to the authors') in terms of the coding efficiency. Moreover, the presented results include many insights: the effectiveness of inter-view cross attention compared to the conventional disparity-based prediction, the advantage of the view-symmetric design over the asymmetric one, and the effectiveness of the joint encoder-decoder training for making the latent representation more efficient.

Weaknesses

- I have a concern for Fig. 4 (c) and (f). The authors state that these graphs are drawn for the selected cameras (C1 and C4). However, most of the methods presented in these graphs take joint decoding or joint coding-decoding scheme, where the information from all the cameras are jointly used. Therefore, it is theoretically impossible/meaningless to define the bitrate only for the selected cameras.

- As the authors admitted, a simple average pooling for merging multi-view information is not sufficient enough; as shown in Table 3, increasing the number of views did not much contribute to the reconstruction quality.


**Summary Of The Paper:**

The authors proposed a learning-based multi-view coding framework comprising of  individual encoding and joint decoding of a set of multi-view images. Under their framework, inter-view coherency can be used for improving the coding performance while eliminating the need of communication among the distributed encoders for individual views. The proposed method was designed to be symmetric: each view is treated equally, and independent of the number of viewpoints. To this end, they incorporated joint context transfer modules into the decoder, where the features from a specific viewpoints are enhanced by those from the other viewpoints via the multi-head cross attention module. The experimental results include extensive comparisons with previous works in terms of the coding performance and computational efficiency, ablation studies for verifying their distributed framework and joint decoding design, and subjective evaluations.

**Summary Of The Review:**

The paper is well written, the authors' proposal has sufficient novelty, and experimental results are extensive and informative for the community. However, I have several concerns for the correctness (Fig. 4 (c) and (f)) and clarity. I would like to recommend acceptance for this work, but not strongly so.

---

> ### Author Response · Authors · 2022-11-08
> **Response to reviewer 8AGJ**
>
> Many thanks for your valuable comments and the recognition of our contributions. We hope the following responses could address your concerns.
>
> **Comment 1: I have a concern for Fig. 4 (c) and (f). The authors state that these graphs are drawn for the selected cameras (C1 and C4). However, most of the methods presented in these graphs take joint decoding or joint coding-decoding scheme, where the information from all the cameras are jointly used. Therefore, it is theoretically impossible/meaningless to define the bitrate only for the selected cameras.**
>
> Thank you for this comment. The reviewer appeared to misunderstand the experimental results in Fig. 4 (c) and (f), where we only jointly compress the images from two cameras C1 and C4 without using additional information from other cameras. We have clarified our settings to avoid ambiguity in the revised paper.
>
> **Comment 2: As the authors admitted, a simple average pooling for merging multi-view information is not sufficient enough; as shown in Table 3, increasing the number of views did not much contribute to the reconstruction quality.**
>
> We agree with the reviewer that this is a limitation in our work. In the future, we will try more complex aggregation approaches such as rank-based average pooling [1], attention-weighted pooling [2], and spectral pooling [3] to further improve the compression performance.
> It is worth investigating how to more effectively incorporate different view information to generate a better inter-view context.
>
> [1] Shi, Zenglin, Yangdong Ye, and Yunpeng Wu. "Rank-based pooling for deep convolutional neural networks." Neural Networks 83 (2016): 21-31.
>
> [2] Dippel, Jonas, Steffen Vogler, and Johannes Höhne. "Towards fine-grained visual representations by combining contrastive learning with image reconstruction and attention-weighted pooling." arXiv preprint arXiv:2104.04323 (2021).
>
> [3] Rippel, Oren, Jasper Snoek, and Ryan P. Adams. "Spectral representations for convolutional neural networks." Advances in neural information processing systems 28 (2015).
>
> **Comment 3: The meaning of "the fast variant" of the proposed method is not clear enough.**
>
> The fast variant of the proposed method uses the parallelization-friendly checkerboard entropy model in [4] instead of the serial auto-regressive entropy model. The checkerboard entropy model allows parallelization encoding and decoding, which can achieve faster inference speed. Specifically, the latent representation $\boldsymbol{\hat{y}}$ is split to anchors $\boldsymbol{\hat{y}}\_{1}$ and non-anchors $\boldsymbol{\hat{y}}\_{2}$ in spatial dimension. When encoding/decoding the latent representation $\boldsymbol{\hat{y}}$, the entropy parameters of anchors $\boldsymbol{\hat{y}}\_{1}$ are first calculated by only using the side information $\boldsymbol{\hat{z}}$. After compressing/decompressing $\boldsymbol{\hat{y}}\_{1}$, the entropy parameters of non-anchors $\boldsymbol{\hat{y}}\_{2}$ are calculated from the side information $\boldsymbol{\hat{z}}$ and context $\boldsymbol{\hat{y}}\_{1}$. We have added more details about the fast variant of the proposed method in the revised paper.
>
> [4] Dailan He, Yaoyan Zheng, Baocheng Sun, Yan Wang, and Hongwei Qin. Checkerboard context model for efficient learned image compression. In Proceedings of the IEEE/CVF Conference on Computer Vision and Pattern Recognition, pp. 14771–14780, 2021.
>
> **Comment 4: It would be better if this categorization was clearly presented in Fig. 4 and Table 1 as well.**
>
> We have added the corresponding categorization for the competing baselines in Figure 4 and Table 1.
>
> **Comment 5: More implementation details for the ablation study is helpful for readers.**
>
> Based on the proposed method shown in Figure 2, we insert two joint context transfer (JCT) modules after the second and fourth convolution layers at the encoder to implement the _Joint Enc-Dec_, thereby allowing both the encoder and the decoder to access the inter-view context. For the _Sep Enc-Dec_, the JCT modules at the decoder are removed, which is equivalent to single image compression. These models are trained based on the InStereo2K dataset by using the same training scheme as in our LDMIC, which is illustrated in Section 4.1. Specifically, the model is trained from scratch for 400 epochs on InStereo2K by using Adam optimizer, in which the batch size is taken as 8. The learning rate is initially set as $10^{-4}$ and decreased by a factor of 2 every 100 epochs until it reaches 400 epochs.
> For the _W/O Joint Training_ case, we fix the pre-trained encoder and entropy model on the _Sep Enc-Dec_ and only train the joint decoder on the InStereo2K dataset, which follows the same training procedure as in our proposed method. These experimental details have be deferred to Appendix 6.5.

---

> ### Author Response · Authors · 2022-11-16
> **Response to reviewer 8AGJ**
>
> Many thanks for your valuable comments. The manuscript has been revised and the main changes are highlighted in blue. If you have any questions, please do not hesitate to mention it to us. We are more than happy to address your concerns.

---

### Official Review · Reviewer_XGBn · 2022-10-24

**Confidence:** 2
**Correctness:** 3
**Technical Novelty And Significance:** 2
**Empirical Novelty And Significance:** 2
**Recommendation:** 6

**Clarity, Quality, Novelty And Reproducibility:**

This paper reads well and is easy to follow. However, the technical novelty seems to be limited. As the proposed architecture is similar to existing methods such as VQVAE and VQGAN, I wonder why the authors want to leverage this architecture to tackle this task. Also, I encourage the authors to provide the implementation and the pre-trained model for reproducibility.

**Strength And Weaknesses:**

- Strengths:
  - This paper is well-written and easy to follow.
  - The proposed method performs favorably against existing image compression methods.
  - The proposed protocol for the distributed system is interesting and useful.
- Weaknesses:
  - Since this task is mainly about compressing multiview images, there should be some priors that can be leveraged. Such as epipolar constraints. However, I do not see any special design of the proposed method in addition to handling multi-frame. Is there any reason for not using these priors to encode multiview images better?
  - I do not see many technical novelties in this paper. The proposed architecture is similar to VQVAE and VQGAN. The authors do not specifically point out how you integrate the existing architecture into this task and why does it help to get better compression results?

**Summary Of The Paper:**

This paper proposed a learning-based approach for multi-view image coding. Specifically, the proposed method decouples the inter-view operations during the encoding and thus is suitable for the distributed system. Thorough evaluations demonstrate the effectiveness of the proposed method in terms of PSNR and SSIM.

**Summary Of The Review:**

This paper achieves the best performance on the task of multiview image coding. However, I do not see many technical novelties in the method or architecture design. The most exciting part is that one can use the proposed method for the distributed system. I lean toward accepting this paper but rejecting it is also acceptable.

---

> ### Author Response · Authors · 2022-11-08
> **Response to reviewer XGBn**
>
> **Comment 2: I do not see many technical novelties in this paper. The proposed architecture is similar to VQVAE and VQGAN. The authors do not specifically point out how you integrate the existing architecture into this task and why does it help to get better compression results?**
>
> With regard to the novelty and technical contribution of our work, the statements in the previous manuscript may not be clear. Thus, we clarify this point here and revise the manuscript accordingly.
>
> - To the best of our knowledge, we are the first work to develop a novel deep learning-based _view-symmetric_ framework for multi-view image coding. It decouples the inter-view operations at the encoder, which is highly desirable for distributed camera systems.
>
> - We present a joint context transfer module at the decoder to explicitly capture inter-view correlations for generating more informative representations and propose an end-to-end encoder-decoder training strategy to implicitly make the latent representation more compact.
>
> - Extensive experimental results show that our proposed framework is the first distributed codec to achieve a coding performance comparable to the state-of-the-art joint encoding-decoding schemes. For example, when compared with SASIC, our method and the fast variant reduce around 21.86\% and 9.85\% bits in terms of PSNR, respectively. It verifies the effectiveness of inter-view cross-attention mechanism compared to the conventional disparity-based prediction. Besides, compared with the _asymmetric-based_ coding framework NDIC, our proposed method achieves higher coding efficiency. This demonstrates the advantage of the view-symmetric design.
>
> **To address the reviewer's concern about the novelty, we also discuss the difference between the proposed method and the VQVAE and VQGAN methods in the following.**
>
> - Application: Both VQVAE and VQGAN aim at image generation and are suitable for single image compression, while our proposed framework is designed for multi-view image compression.
> - Technical route: The main components of VQVAE and VQGAN are encoder, vector quantizer and decoder. They are built upon variational autonencoders through the use of a vector-quantized, discrete latent space. Whereas, our proposed method is developed from the popular learning-based image coders [2-4] following the transform coding paradigm from traditional image compression methods. Typically, transform coding methods are composed of three components, i.e., transform, quantizer, and entropy codec. They operate by linearly or non-linearly transforming the data vector into a suitable continuous-valued representation, quantizing its elements independently, and then encoding the resulting discrete representation using a lossless entropy codec. The learning-based image coders achieve high compression efficiency by implementing the nonlinear transformations and an entropy model using neural networks in an end-to-end way.
>
> **We further illustrate why we can achieve better results on top of learning-based single image compression.**
>
> Firstly, our proposed method exploits the correlation of different views as much as possible to achieve higher reconstruction quality by performing a joint nonlinear transformation operation at the decoder side. Specifically, we introduce a joint decoder with cross-attention mechanism to better utilize inter-view correlations. Our ablation study in Figure 5 demonstrates the effectiveness of the joint decoder. Secondly, as shown in Figure 6, with the help of the joint training strategy, our proposed framework makes the latent representation more compact.
>
> To sum up, based on the single image compression model, the proposed method not only uses the joint context transfer module to explicitly exploit inter-view correlations for generating more informative representations, but also utilizes the end-to-end training strategy to implicitly facilitates the encoder to learn to remove the partial inter-view redundancy.
>
> [2] Johannes Balle, Valero Laparra, and Eero P Simoncelli. End-to-end optimized image compression. In International Conference on Learning Representations, 2017.
>
> [3] Johannes Balle, David Minnen, Saurabh Singh, Sung Jin Hwang, and Nick Johnston. Variational image compression with a scale hyperprior. In International Conference on Learning Representations, 2018.
>
> [4] David Minnen, Johannes Balle, and George D Toderici. Joint autoregressive and hierarchical priors for learned image compression. Advances in neural information processing systems, 31, 2018.
>
> **Comment 3: I encourage the authors to provide the implementation and the pre-trained model for reproducibility.**
>
> Thank you for this advice, we will make our code public, which is coming soon.

---

> ### Author Response · Authors · 2022-11-08
> **Response to reviewer XGBn**
>
> Many thanks for your valuable comments and the recognition of our contributions. We hope the following responses could address your concerns.
>
> **Comment 1: I do not see any special design of the proposed method in addition to handling multi-frame. Is there any reason for not using these priors to encode multiview images better?**
>
> We agree with the reviewer that leveraging priors (e.g., epipolar constraints) in image encoding could improve the compression ratio. However, applying this epipolar geometric constraint requires the internal and external parameters of the camera in advance such as camera locations, orientations, and camera matrices, which may be difficult to obtain in some applications. For example, in order to avoid leaking the location and trajectory of individuals in autonomous driving, the specific location information of the camera is usually not expected to be perceived by other vehicles or infrastructure [1].
>
> In this paper, we proposed a general approach that does not rely on priors, where we employed a cross-attention mechanism to exploit the global content dependencies between different views at the decoder, which is agnostic to the geometric relationships between different views. Thus, this method can be used in arbitrary multi-camera systems with overlapping fields of view.
>
> [1] Z. Xiong, Z. Cai, Q. Han, A. Alrawais and W. Li, "ADGAN: Protect Your Location Privacy in Camera Data of Auto-Driving Vehicles," In IEEE Transactions on Industrial Informatics, vol. 17, no. 9, pp. 6200-6210, Sept. 2021.

---

> ### Author Response · Authors · 2022-11-16
> **Response to reviewer XGBn**
>
> Many thanks for your valuable comments. The manuscript has been revised and the main changes are highlighted in blue. If you have any questions, please do not hesitate to mention it to us. We are more than happy to address your concerns.

---

### Official Review · Reviewer_11ke · 2022-10-27

**Confidence:** 3
**Clarity, Quality, Novelty And Reproducibility:** I believe that this paper is of high …
**Correctness:** 4
**Technical Novelty And Significance:** 3
**Empirical Novelty And Significance:** 3
**Recommendation:** 8

**Strength And Weaknesses:**

Strength:
+ The paper is well written and easy to follow
+ The results are good compared with SOTA
+ The experiments are comprehensive and well evaluate the performance on different datasets

Weakness:
- It will be great if there's a short discuss of failure cases.

**Summary Of The Paper:**

This paper proposed a novel deep learning based framework to effectively encode the distributed multi-view images. Unlike previous multi-view image coding (MIC) architectures that utilized a shared encoder to explore the inter-view redundancy, the proposed method moved this part to decoder and successfully reduced the computational costs in MIC. The experiments demonstrate the robustness and efficiency of proposed method.

**Summary Of The Review:**

In summary, this paper proposed a novel coding algorithm that can decouple the encoding of images acquired by distributed cameras. The method seems to effective and the results are good. The paper is well written and the comparison experiments are comprehensive.

---

> ### Author Response · Authors · 2022-11-08
> **Response to reviewer 11ke**
>
> **Comment: It will be great if there is a short discussion of failure cases.**
>
> Many thanks for your valuable comment and the recognition of our contributions.
> Our proposed method is designed for multi-view images with overlapping regions, and the performance may be degraded in other settings. For example, if the proposed method is applied to two images $A$ and $B$ with almost no correlation, it may result in low rate-distortion performance. Specifically, when decoding image $A$, the cross-attention mechanism may introduce the interference information from image $B$ during joint decoding, which would lead to low reconstruction quality. Therefore, our proposed method may not achieve the desired performance in this case. In other words, it is important to investigate the particular application scenario and camera settings to apply our proposed method.

---

> ### Author Response · Authors · 2022-11-16
> **Response to reviewer 11ke**
>
> Many thanks for your valuable comments. The manuscript has been revised and the main changes are highlighted in blue. If you have any questions, please do not hesitate to mention it to us. We are more than happy to address your concerns.

---

### Author Response · Authors · 2022-11-16
**General Response**

We are very thankful for the area chair to coordinate the review of our manuscript and grateful to the reviewers for their valuable feedback. We very much appreciate the assessment of work as **novel (Reviewer 11ke, 8AGJ)**, **interesting (Reviewer XGBn)**, and **effective (Review 11ke, XGBn, 8AGJ, WX5p)**.

The authors have provided point-to-point responses to the comments raised by the reviewers. A summary of them is as follows:

1. Raised by **Reviewer XGBn** and **Reviewer WX5p**'s concern about the technical novelty of the proposed method, we have revised the manuscript to point out that, to the best of our knowledge, this is the first work developing a deep learning-based distributed view-symmetric framework for multi-view image coding. This coding scheme not only decouples the inter-view operations at the encoder but also achieves a comparable compression ratio to the SOTA joint encoding-decoding schemes.

2. Following the advice of **Reviewer 8AGJ** and **Reviewer WX5p**, we have provided more descriptions of the fast variant of the proposed method and added more details about the experimental setups.

**The authors would like to kindly remind the reviewers to take a look at the responses and see whether the raised concerns have been well addressed. Thank you for your help and expertise! We look forward to hearing from you again.**

---

### Decision · Program_Chairs · 2023-01-20

**Decision:**

Accept: poster

**Justification For Why Not Higher Score:**

There is still room for improvement for this paper, including better motivation and broadly exploring the solution space.

**Justification For Why Not Lower Score:**

The paper introduced a novel application/setup for multi-view image coding, which is a lot more practical & useful than the setup before.

**Metareview: Summary, Strengths And Weaknesses:**

The paper proposed a learning-based distributed multi-view image coding (LDMIC) framework to encode multi-view images without any geometric assumptions by means of decoupling the inter-view operations at the encoder. A join context transfer module at the decoder was used to explicitly capture inter-view correlations. The paper outperformed the latest asymmetric-based coding framework.

All the reviewers are positive about the paper: the paper is well written, it’s the first work to develop a view-symmetric framework with deep learning, and the paper has achieved higher coding efficiency compared with the view-asymmetric coding methods. The reviewers also had concerns that the paper lacks novelty as many existing models such as the encoding-decoding framework have been used in the paper. The authors addressed many of the concerns clearly in the rebuttal, but the reviewers didn’t respond to the rebuttal.

The AC sees the value of the paper: it’s a novel setup that would make the multi-view image coding a lot more practical and useful as camera calibration is no longer needed. However, the paper should be motivated better. The authors need to elaborate more on “In the fields of autonomous driving, virtual reality and video surveillance, multi-view image coding has become one of the most critical techniques”. It’s unclear how the proposed method can directly address the downstream tasks in these areas, so please clarify.


**Note From Pc:**

if the above contains the word "oral" or "spotlight" please see: "oral" presentation means -> notable-top-5% and "spotlight" means -> notable-top-25%. As stated in our emails, we are disassociating presentation type from AC recommendations